# A Novel Photoluminescent Ag/Cu Cluster Exhibits a Chromic Photoluminescence Response towards Volatile Organic Vapors

**DOI:** 10.3390/molecules28031257

**Published:** 2023-01-27

**Authors:** Wei Yang, Shengnan Hu, Yuwei Wang, Sisi Yan, Xiang-Qian Cao, Hong-Xi Li, David James Young, Zhi-Gang Ren

**Affiliations:** 1Suzhou Key Laboratory of Novel Semiconductor-Optoelectronics Materials and Devices, College of Chemistry, Chemical Engineering and Materials Science, Soochow University, Suzhou 215123, China; 2Faculty of Food Science and Technology, Suzhou Polytechnic Institute of Agriculture, Suzhou 215008, China; 3College of Engineering, Information Technology and Environment, Charles Darwin University, Darwin 0909, NT, Australia

**Keywords:** photoluminescence material, vapor-chromic response, Ag/Cu complex, volatile organic vapor (VOC) detection

## Abstract

A new Ag/Cu bimetallic cluster [Ag_10_Cu_6_(bdppthi)_2_(C≡CPh)_12_(EtOH)_2_](ClO_4_)_4_ (**1**, bdppthi = *N*,*N′*-bis(diphenylphosphanylmethyl)-tetrahydroimidazole) exhibited strong phosphorescent (PL) emission at 644 nm upon excitation at 400 nm. Removal of the coordinated EtOH molecules in **1** resulted in derivative **1a**, which exhibited significant red-shifted emission at 678 nm. The structure and PL of **1** was restored on exposure to EtOH vapor. Cluster **1a** also exhibited a vapor-chromic PL response towards other common organic solvent vapors including acetone, MeOH and MeCN. A PMMA film of **1a** was developed as a reusable visible sensor for MeCN.

## 1. Introduction

Volatile organic compounds (VOCs) are hazardous air pollutants. Some coordination compounds can serve as photoluminescent (PL) probes for VOCs with a rapid and reversible switching or chromic response [1,2,3,4,5]. Coinage metal complexes have been developed with useful photophysical properties such as sensitive and selective emission color changes under external stimuli [6,7,8,9,10,11,12,13]. Bimetallic Ag/Cu complexes have attracted considerable attention in this respect due to their low cost and rich luminescence behaviors. For example, Chen et al. have reported an Ag(I)-Cu(I) complex showing reversible vapor-chromic phosphorescence, with the emission changing from bright yellow to green in response to THF or CHCl_3_ [14].

Vapor-chromic UV and PL responses of coordination compounds originate mostly from intramolecular structural distortion, such as the formation/disruption of metal–solvent bonds [15], molecular deformation [16,17] or conformational transformation [18,19]. The PL vapor-chromic response of metal complexes towards VOCs can be attributed to the interactions between the metal or their ligands and the VOC. In some cases, the small VOC molecules enter the lattice voids but do not participate in coordination bonds. There are only weak interactions, such as H-bonding, van der Waal’s forces or C-H···π or π···π interactions between these ‘free solvent molecules’ and the main structure that partly affects the complex’s energy levels [20,21,22,23,24,25,26,27,28]. In other cases, the VOC molecules coordinate with the metal ions to form metal–solvent bonds, which can significantly change the emission color or intensity [29,30,31,32]. For instance, Wang and co-workers have reported that the PL of an Au/Ag cluster reversibly shifted between green and yellow when the weakly ligated methanol molecules were removed or re-introduced [33].

Recently, we found that the strong emission of a Cu/Ag cluster [Ag_10_Cu_6_(bdppthi)_2_(C≡CPh)_12_(MeOH)_2_(H_2_O)](ClO_4_)_4_ (**2**, bdppthi = *N,N′*-bis(diphenylphosphanylmethyl)-tetrahydroimidazole) could be quenched by NH_3_, which enabled its use as a rapid, reversible and visual sensor [34]. During this quenching, the Ag_10_Cu_6_ cluster remained coordinated with MeOH and H_2_O molecules, while the NH_3_ only interacted with the MeOH ligands. Hence, we expected that the PL of this cluster would change when the MeOH and H_2_O molecules were removed or replaced by other small organic molecules, such as VOCs, that could coordinate with the Ag/Cu cluster core directly. We therefore prepared a new cluster [Ag_10_Cu_6_(bdppthi)_2_(C≡CPh)_12_(EtOH)_2_](ClO_4_)_4_ (**1**), in which the solvates were replaced with EtOH. The luminescence of **1** significantly red-shifted on elimination of EtOH to produce **1a**, and was immediately restored upon exposure to EtOH. Other VOC vapors also resulted in a vapor-chromic PL response.

## 2. Results and Discussion

### 2.1. Synthesis and Characterization

#### 2.1.1. Synthesis of **1**

The reaction of *N*,*N′*-bis(diphenylphosphanylmethyl)ethylene diamine (bdppeda), [Cu(MeCN)_4_](ClO_4_) and AgC≡CPh (molar ratio 1:4:4) in CH_2_Cl_2_/EtOH (Figure 1) produced bimetallic cluster **1** in 73% yield. This methodology was similar to that used in the synthesis of compound **2,** except for the solvent [34] and that the ligand bdppthi was generated in situ [35].

#### 2.1.2. Single Crystal Structure of **1**·2CH_2_Cl_2_

Single-crystal X-ray diffraction (SCXRD) of **1**·2CH_2_Cl_2_ revealed that it crystallized in the monoclinic system *P*2_1_/n space group. The asymmetric unit contained one [Ag_10_Cu_6_(bdppthi)_2_(C≡CPh)_12_(EtOH)_2_)]^4+^ tetracation, four ClO_4_^−^ anions and two CH_2_Cl_2_ solvent molecules. As shown in Figure 1, the Ag_10_Cu_6_ cluster core may be viewed as two smaller Ag_6_Cu_3_ units joined by sharing two Ag(I) cations. The two bdppthi ligands stabilized and connected the two Ag_6_Cu_2_ units through four Ag-P and four Cu-N bonds. Each Cu(I) atom was coordinated oppositely by two C≡CPh anions. The average Ag-P, Cu-N and Cu-C bond lengths were 2.383(3), 2.271(9) and 1.915(13) Å, respectively. The Ag_10_Cu_6_ cores in compounds **1** and **2** were somewhat analogous. The Ag-Ag and Ag-Cu distances were in the range of 2.829–3.310 Å and 2.701–3.180 Å, respectively, indicating the presence of substantial metallophilic interactions [36]. Nevertheless, the metal–metal distances in **1** were slightly longer than those in **2**. The smallest Cu-Cu distance of 2.969(2) Å (between Cu2 and Cu3) was larger than the sum of the van der Waals radii of the two Cu atoms (2.80 Å), which ruled out the existence of Cu-Cu interactions and revealed that the Ag_10_Cu_6_ cores in **1** and **2** were different. The two Cu(I) atoms at the end of the Ag_10_Cu_6_ cluster core were further coordinated with the O atoms of two EtOH molecules (Cu1-O1, 2.043(8) Å; Cu4-O2, 2.031(9) Å). These two Cu-O (EtOH) bond lengths were slightly shorter than the Cu-O (MeOH) (2.1134(1) and 2.2071(1) Å) and Cu-O (H_2_O) (2.1428(1) Å) interactions in **2**. These differences in bond lengths demonstrated that the metallophilic interactions in the Ag_10_Cu_6_ core were weakened when EtOH molecules were closely coordinated to the end Cu ions. In addition, there were weak interactions between two of the ClO_4_^−^ anions and these two Cu atoms: (Cu1-O5, 2.758(1) Å; Cu4-O9, 2.715(1) Å), while the other two ClO_4_^−^ anions remained free.

#### 2.1.3. Characterization of **1**

The CH_2_Cl_2_ molecules in **1**·2CH_2_Cl_2_ readily escaped in air as evidenced by an absence of interaction in the solid-state structure. Thus, all other characterizations were performed on **1**. This cluster readily dissolved in common organic solvents such as CH_2_Cl_2_, CHCl_3_, acetone, DMSO, DMF and MeCN, but was insoluble in Et_2_O, hexane and H_2_O. The elemental analysis of **1** was consistent with its molecular formula. The powder X-ray diffraction (PXRD) pattern of **1** correlated with the simulated spectra generated from the SCXRD data and was clearly different from that of **2** (Figure 2). The IR spectrum showed characteristic peaks at 2019 cm^−1^ for C≡C, 1082 cm^−1^ for ClO_4_^−^, and at 1483, 1435, 750, 687 and 619 cm^−1^ for the −Ph groups (Appendix A). The positive-ion electrospray ion mass spectrometry (ESI-MS) of **1** (Appendix A) contained peaks attributed to [Ag_2_(bdppthi)(C≡CPh)]^+^ (*m/z* = 785.03) and [Ag(bdppthi)]^+^ (*m*/*z* = 575.09) cations and {[Ag_10_Cu_4_(C≡CPh)_9_(H_2_O)](ClO_4_)_3_+e+H^+^}^2+^ (*m*/*z* = 1278.89) dications, indicating the structure of **1** partly decomposed in the ESI-MS environment.

#### 2.1.4. Interconversions of **1** and **1a**

Thermogravimetric analysis (TGA) of **1** in a N_2_ stream showed a weight loss of 2.2% between 120 and 150 °C, which matched the elimination of the two coordinated EtOH molecules (Calcd 2.24%) (Appendix A). We therefore treated **1** in vacuum by heating at 120 °C for 1 h and obtained its solventless derivative [Ag_10_Cu_6_(bdppthi)_2_(C≡CPh)_12_](ClO_4_)_4_ (**1a**). Cluster **1a** was less crystalline than **1** and not suitable for SCXRD analysis. The weak IR vibration at 3420 cm^–1^ attributed to the −OH group of EtOH in **1** disappeared in **1a** (Appendix A). As shown in Figure 2, the PXRD pattern of **1a** was similar to that of **1**, indicating its cell parameter and major structure remained unchanged. The elimination of some weak peaks might be due to the elimination of the EtOH molecules. When **1a** was exposed to EtOH vapor for several minutes, the PXRD pattern fully recovered, indicating re-coordination of the EtOH molecules. However, when **1a** was treated with MeOH vapor, the PXRD pattern resembled that of compound **2**, indicating a phase-transition. This transition was reversible, so that desolvation of **2** and re-exposure to EtOH produced **1** quantitatively.

### 2.2. Photoluminescent Properties

#### 2.2.1. Photoluminescent Properties of **1**

Upon excitation at 400 nm, crystals of **1** exhibited bright red emission at *λ*_max_ = 644 nm (Figure 3) with a quantum yield (QY) of 10% at ambient temperature. The PL lifetime (*τ*, excited at 373 nm) was 7.48 μs. This relatively long lifetime and large Stokes shift (244 nm) suggested that this PL was a phosphorescent emission and likely arises from a metal cluster-centered triplet excited state modified by metal –metal interactions, mixed with a [C≡CPh→Ag_10_Cu_6_] ^3^LMCT transition [14,34]. The emission was not sensitive to excitation wavelengths, as the spectrum excited at 365 nm was similar to that excited at 400 nm except for a small intensity decay.

#### 2.2.2. Photoluminescent Properties of **1a** and Vapor-Chromic Responses toward Water and VOCs

Solid **1a** emitted at *λ*_max_ = 678 nm when excited at 400 nm (Figure 4). Compared to that of **1**, this emission exhibited a 34 nm red-shift, reduced intensity (QY = 6%), and a slightly prolonged lifetime (*τ* = 8.82 us, excited at 373 nm). The emission of **1a** could be fully restored to that of **1** (644 nm) after exposure to EtOH vapor, and this process showed good repeatability over four cycles (Appendix A). The PXRD of **1a** remained steady during these cycles and for as long as one month later (Appendix A). We believe that during the interconversion of **1** and **1a**, the departure and re-coordination EtOH molecules from the Cu atoms affect the electron density of the Ag_10_Cu_6_ cluster center, which influences the T1→S0 energy band, and therefore the shifting of the emission wavelength and intensity arises.

Complex **1a** was relatively stable toward air and moisture. The emission of **1a** shifted to 718 nm when its powder was immersed in liquid water for 20 h (Appendix A). We suggested this red-shift was caused by the interaction between the Ag_10_Cu_6_ cluster center and water molecules. This interaction was weak and lessened when the content of water was decreased. Therefore, a low-energy shoulder could be observed at the emission curve of **1a**, which indicated that a little portion of **1a** was hydrated by moisture when it was put in open air for days.

The emission of **1a** changed when this cluster was exposed to other VOC vapors. The less-coordinating VOC molecules, such as CH_2_Cl_2_, CHCl_3_ and Et_2_O, caused only minor blue-shifting (*λ*_max_ = 666, 671 and 670 nm, respectively), whereas those VOCs with stronger donor groups, including acetone (C=O), MeOH (-OH) and MeCN (-CN), caused obvious blue-shifted emissions with *λ*_max_ = 658 nm (acetone), 628 nm (MeOH) and 594 nm (MeCN), respectively. The changes in emission wavelength on exposure of **1a** to MeOH and MeCN were 50 nm and 84 nm, respectively, which is visible to the naked eye under 365 nm LED irradiation (Figure 4).

### 2.3. Photoluminescent Probe for the Detection of MeCN

The significant vapor-chromic response of **1a** encouraged us to prepare a PL sensing film by compositing **1a** with PMMA (3% *w*/*w*). This film showed a visible red-to-orange PL change on exposure to a saturated atmosphere of MeCN vapor (Figure 5). The red emission could be recovered upon heating in air at 100 °C. This color interconversion was repeatable, giving a reusable PL probe for the detection of VOCs.

## 3. Materials and Methods

### 3.1. Materials and Measurements

Bdppeda [37,38] and AgC≡CPh [39] were prepared by literature procedures. [Cu(MeCN)_4_](ClO_4_) was commercially available. Elemental analyses (C, H and N) were performed on a Carlo Erba CHNO-S microanalyzer. PXRD measurements were recorded on a Bruker D2 Phaser X-ray diffractometer with a Cu *K*α source (30 kV, 10 mA). IR spectra were obtained on a VERTEX 70 FT-IR spectrometer (4000–500 cm^−1^) with an ATR probe. Thermogravimetric analysis (TGA) was completed on a TA SDT-600 analyzer under an N_2_ atmosphere in the range from room temperature to 900 °C, with a temperature heating rate of 10 °C/min. PL measurements were performed on an Edinburgh FLS1000 spectrophotometer. Positive-ion electrospray ion mass spectrometry (ESI-MS) was recorded on a Bruker microTOF-Q III mass spectrometer using MeOH as the mobile phase.

### 3.2. Synthesis of **1** and **1a**

A mixture containing bdppeda (46 mg, 0.1 mmol), [Cu(MeCN)_4_](ClO_4_) (131 mg, 0.4 mmol), 5 mL of CH_2_Cl_2_ and 5 mL of EtOH was stirred for 0.5 h at ambient temperature. AgC≡CPh (84 mg, 0.4 mmol) was added into the resulting colorless solution and stirred for 3 h. The mixture turned red and was subsequently centrifuged. The supernatant was diffused with Et_2_O and afforded red crystals of **1**·2CH_2_Cl_2_ after 1 day. The CH_2_Cl_2_ solvate escaped quickly in air, leaving **1** within an hour. Yield for **1**: 117 mg (73% based on Ag). Anal. Calcd for C_158_H_132_Ag_10_Cl_4_Cu_6_N_4_O_18_P_4_: C, 46.28; H, 3.24; N, 1.37. Found: C, 45.82; H, 3.34; N, 1.26. IR (ATR, cm^−1^): 3419 (w), 3059 (w), 2019 (m), 1483 (m), 1435 (m), 1082 (vs), 750 (s), 687 (vs), 619(s).

Complex **1** was placed in a vacuum oven at 120 °C for 1 h and produced **1a** on cooling to room temperature. The yield was almost quantitative.

### 3.3. Preparation of **1a**/PMMA

A 9 mg sample of **1a** was carefully ground in a mortar and pestle, dispersed in PMMA/toluene solution (20% *w*/*w*, 1.5 g) and sonicated for 30 min. The mixture was applied to glass slides (3 × 6 cm^2^) and left to dry in air over several hours. The dried film was removed from the slide and cut into small pieces (1.5 × 4 cm^2^).

### 3.4. Single-Crystal Crystallography

A single crystal of **1**·2CH_2_Cl_2_ was collected from the above synthesis. SCXRD analysis was performed on a Bruker D8 Venture diffractometer using graphite-monochromated Ga Kα (*λ* = 1.34138 Å) radiation at 120 K. All data were integrated with Bruker SAINT and a multi-scan absorption correction was applied. The structure was solved by direct methods using SHELXS 2016/6 (Sheldrick, 2016) and refined by full-matrix least-squares methods against *F*^2^ by SHELXL-2016/6 (Sheldrick, 2016) [40]. All non-hydrogen atoms were refined anisotropically. The hydrogen atoms of the -OH groups of EtOH were first located from a Fourier map and then refined to ride on the O atoms. All other hydrogen atoms were added in idealized positions and constrained to ride on their parent atoms. The data were deposited to the Cambridge Crystallographic Data Centre (CCDC number 2226387). A summary of the key crystallographic data is given in Table 1. Selected bond lengths and angles are listed in Appendix A.

## 4. Conclusions

An Ag_10_Cu_6_ cluster **1** stabilized by PNNP ligand bdppthi and C≡CPh anions was prepared. The two Cu ends of the Ag_10_Cu_6_ core were coordinated with EtOH molecules. Removing these solvates from **1** by vacuum heating produced **1a**, which could be restored to **1** by exposure to EtOH vapor. The maximum PL emission of **1** at 644 nm shifted to 678 nm when converted to **1a**. This reversible vapor-chromic response also occurred with other VOCs, particularly those with polar functional groups such as MeCN and MeOH, which exhibited the largest blue-shift of emissions up to 50 and 84 nm, respectively. A reusable chromic PL probe containing 3% (*w*/*w*) **1a** in PMMA exhibited a visible red-to-orange emission change when exposed to MeCN vapor.

## Figures and Tables

**Figure 1 molecules-28-01257-f001:**
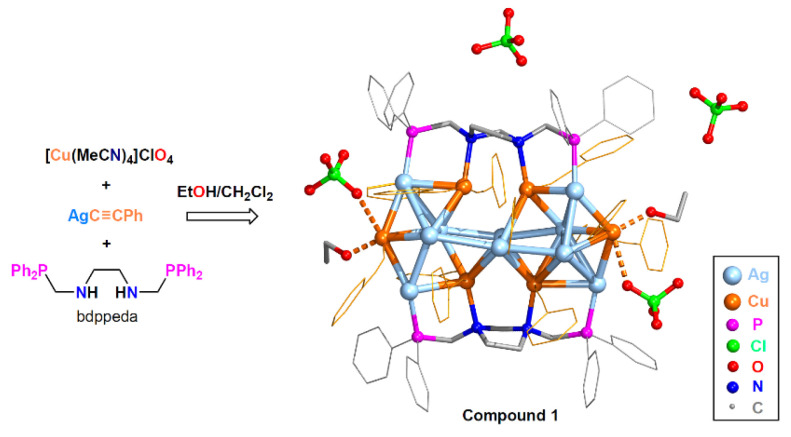
Synthesis and structure of **1**. The phenyl rings of -PPh_2_ groups and C≡CPh groups are plotted as gray and yellow hexagons. CH_2_Cl_2_ solvent molecules are omitted for clarity.

**Figure 2 molecules-28-01257-f002:**
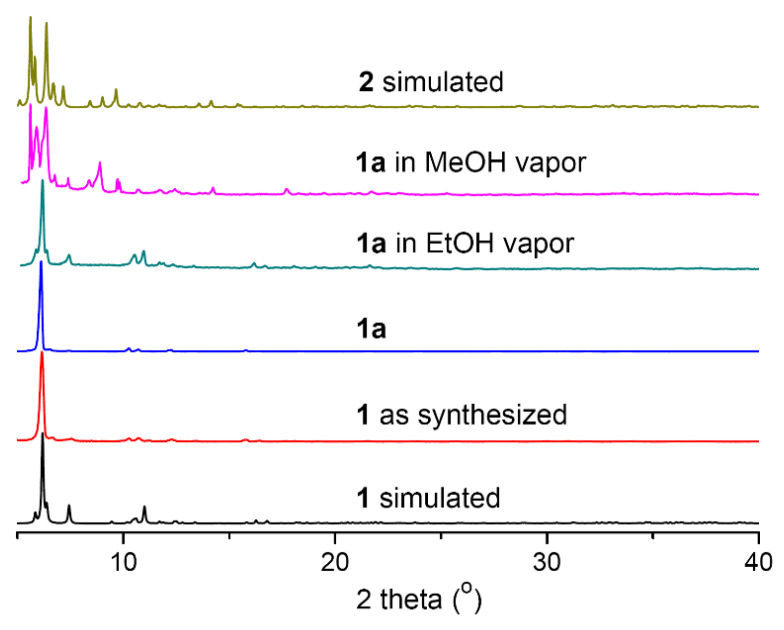
PXRD patterns of compound **1**, **1a** in air, **1a** in EtOH and MeOH vapors, and the simulated patterns of **1** and **2** from the SCXRD data.

**Figure 3 molecules-28-01257-f003:**
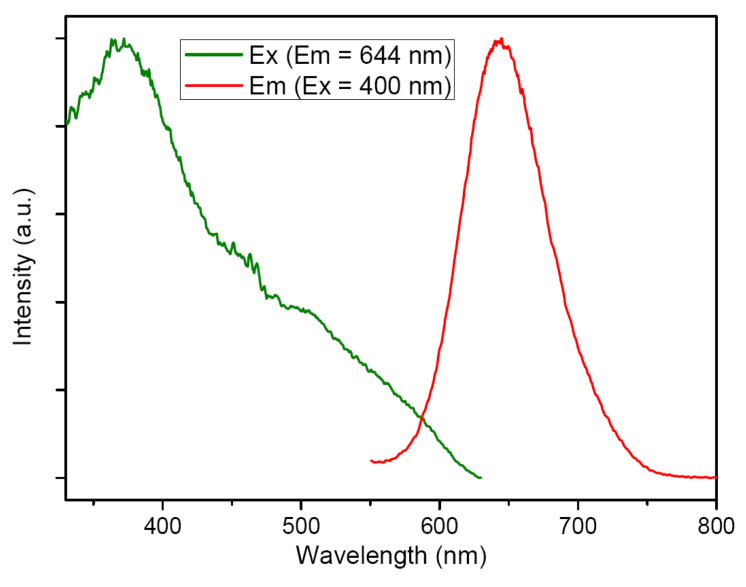
Excitation and emission spectra of **1** at ambient temperature.

**Figure 4 molecules-28-01257-f004:**
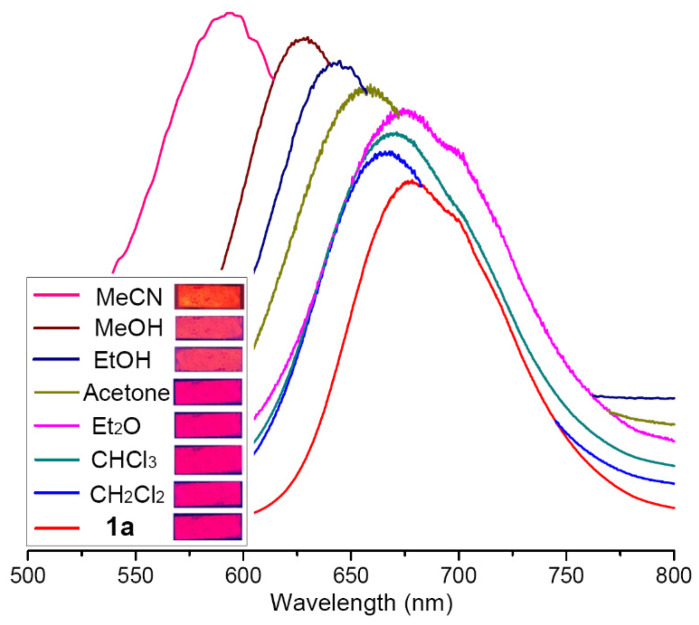
Emission spectra of **1a** and **1a** on exposure to different VOC vapors upon excitation at 400 nm. Photos of the powdered **1a** and **1a** (coating on a quartz slice) on exposure to different VOC vapors under 365 nm excitation (inset).

**Figure 5 molecules-28-01257-f005:**
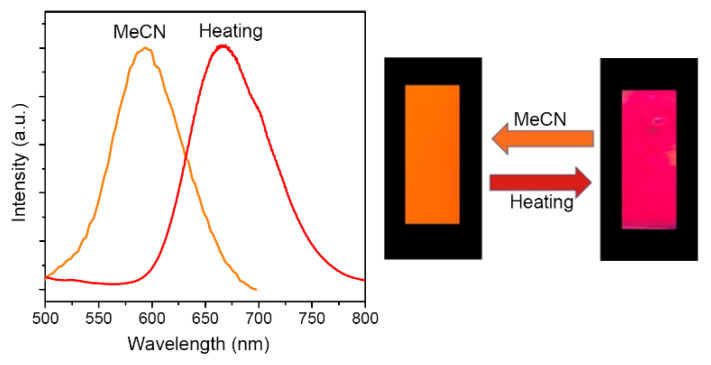
Emission spectra (left, Ex = 400 nm) and photos (right, under 365 nm UV light) of **1a**/PMMA film in MeCN vapor and heating to 100 °C.

**Table 1 molecules-28-01257-t001:** Selected crystallographic data and refinement parameters for **1**·2CH_2_Cl_2_.

Compound	1·2CH_2_Cl_2_
Empirical formula	C_160_H_136_Ag_10_Cl_8_Cu_6_N_4_O_18_P_4_
Formula weight	4270.14
Crystal system	monoclinic
Space group	*P*21/*n*
*a*/Å	30.834(2)
*b*/Å	16.5897(11)
*c*/Å	33.5688(19)
*β*/°	116.652(2)
*V*/Å3	15,346.8(17)
*Z*	4
*ρ*_calc_/g·cm^−3^	1.848
*μ*/mm^−1^	12.665
*F*(000)	8432
*θ*_max_/°	56.966
No. of reflections measured	345,696
No. of independent reflections	31,366 (*R*_int_ = 0.1724)
Data/restraints/parameters	31,366/79/1867
*R*_1_ [*I* > 2.00 σ(*I*)] ^a^	0.1049
w*R_2_* (all reflections)	0.3419
Goodness of fit	1.209

^a^*R*_1_ = Σ||Fo|–|Fc||/Σ|Fo|, w*R*_2_ = {Σw(Fo^2^ – Fc^2^)^2^/Σw(Fo^2^)^2^}^1/2^, GOF = {Σw((Fo^2^ – Fc^2^)^2^)/(*n* – *p*)}^1/2^, where *n* = number of reflections and *p* = total number of parameters refined.

## Data Availability

The crystallographic data are available from the Cambridge Crystallographic Data Centre (CCDC). Other data not presented in the Appendix A are available on request from the corresponding author.

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
