# Peer review of "A Novel Photoluminescent Ag/Cu Cluster Exhibits a Chromic Photoluminescence Response towards Volatile Organic Vapors"

_molecules, 2023, doi:10.3390/molecules28031257_

Round 1
Reviewer 1 Report
The reviewed manuscript describes a new Ag10Cu6 cluster supported by phenylethynide and diphosphine ligands. The cluster complex was structurally and spectrally characterized, as well as investigated in terms of solid-state luminescence. Moreover, it revealed a very interesting vapochromilc luminescence associated with reversable coordination-discoordination of EtOH molecule. Although a very similar cluster bearing MeOH co-ligand was already reported, I believe that this work deserves publishing in this journal. The manuscript itself is well-written and well-designed. Thus, I recommend acceptance of this manuscript after addressing the following concerns:
1. What is a possible origin of a low-energy shoulder which is presented in some of the emission curves plotted in Figure 4?
2. The low-intense XRD peaks are almost invisible in Figure S5.
3. Is the emission of cluster 1 sensitive to excitation wavelengths?
4. Please note that the phrase “deprotonated C≡CPh anions” is not correct.
5. Finally, I guess kindly referencing some other phosphine-based Cu/Ag/Au complexes/clusters, whose emission is also changeable via (de)coordination of solvent molecules, e.g. DOI: 10.1021/acs.inorgchem.2c02506, 10.1021/acs.inorgchem.0c01171, 10.1021/acs.inorgchem.5b00239, 10.1021/acs.inorgchem.2c01474.
Author Response
- What is a possible origin of a low-energy shoulder which is presented in some of the emission curves plotted in Figure 4?
Answer: As indicated by Question 2 of Reviewer 2, we investigated the influence of water toward the PL of 1a. Solid 1a was relatively stable in air, but its emission could shift to 718 nm when it was immersed in water for 20 hours. It seems like that the low-energy shoulder in the curves of 1a and 1a in CH2Cl2, CHCl3, Et2O and acetone was attributed to the partly hydration of 1 by moisture in air. We amended this explanation in the revised manuscript as “Therefore, a low-energy shoulder could be observed at the emission curve of 1a, which indicated that a little portion of 1a was hydrated by moisture when it was put in open air for days.” (Page 4, Lines 155-157).
- The low-intense XRD peaks are almost invisible in Figure S5.
Answer: We re-plotted Figure S5 to make the low-intense XRD peaks clearer in the revised manuscript.
- Is the emission of cluster 1 sensitive to excitation wavelengths?
Answer: The emission of 1 is not sensitive to excitation wavelengths when we changed the excitation wavelength from 400 to 365 nm. This is amended in the revised manuscript as “The emission was not sensitive to excitation wavelengths as the spectrum excited at 365 nm was similar to that excited at 400 nm except for a small intensity decay.” (Page 4, Lines 136-138).
- Please note that the phrase “deprotonated C≡CPh anions” is not correct.
Answer: Thank you for this reminding. We deleted the word “deprotonated” in the revised manuscript.
- Finally, I guess kindly referencing some other phosphine-based Cu/Ag/Au complexes/clusters, whose emission is also changeable via(de)coordination of solvent molecules, e.g. DOI: 10.1021/acs.inorgchem.2c02506, 10.1021/acs.inorgchem.0c01171, 10.1021/acs.inorgchem.5b00239, 10.1021/acs.inorgchem.2c01474.
Answer: We greatly thank you for supplying these important references, and have addressed them in the revised manuscript as refs. 6, 10, 12 and 13.
Reviewer 2 Report
The paper reports a AgCu cluster and its luminescence response towards various solvents. The paper is interesting in terms of luminescence sensing. The authors have recently reported a similar compound differing only in the type of terminal solvents. Thus, the reported cluster is not really novel. The novelty of the work lies in the sensing properties and thus, the authors should include some additional data concerning the sensing studies. Some suggestions-questions for the authors:
1. Can compound 2 in its desolvated form (after removing coordinated MeOH solvent) show similar response to MeCN and other VOCs such as desolvated compound 1? With other words, why someone needs to prepare the EtOH-containing cluster and not use the published compound 2 for VOCs sensing? The authors need to perform a comparison of the sensing properties of 1 and 2 (i.e. to provide the MeCN, CH2Cl2 etc sensing property of the published compound 2 after its desolvation).
2. What about the sensing property of compound 1 towards water? Is the desolvated compound 1 stable to air? Which is the luminescence of desolvated compound 1 after its exposure to air and humidity?
Author Response
- Can compound 2 in its desolvated form (after removing coordinated MeOH solvent) show similar response to MeCN and other VOCs such as desolvated compound 1? With other words, why someone needs to prepare the EtOH-containing cluster and not use the published compound 2 for VOCs sensing? The authors need to perform a comparison of the sensing properties of 1 and 2 (i.e. to provide the MeCN, CH2Cl2 etc sensing property of the published compound 2 after its desolvation).
Answer: Yes, as we indicated that, the desolvated form of compound 2 showed same PXRD pattern to 1a. In fact, it is the other route to prepare 1a. Compound 1a prepared from the two methods exhibited same response toward the VOCs mentioned in this manuscript.
After we finished our previous research about compound 2 (ref. 34), we soon found that the desolvation of 2 showed these similar responses toward VOCs. Whereas, the PXRD pattern of these desolvated solids were different form that of 2, and were less crystalline. We tried many attempts to measure its structure, and finally found that the reactions in EtOH could resulted the single crystals of 1, whose PXRD was correlated with the desolvated samples. Considering that the single crystal structures and PXRD patterns of 1 and 2 were different, and the PXRD pattern of 1a was similar to that of 1, we described compound 1 and a new compound and was the precursor of 1a in this manuscript. Nevertheless, we statemented this fact in the manuscript in Page 4, Lines 123-126 as “However, when 1a was treated with MeOH vapor, the PXRD pattern resembled that of compound 2, indicating a phase-transition. This transition was reversible so that desolvation of 2 and re-exposure to EtOH produced 1 quantitatively.”
- What about the sensing property of compound 1 towards water? Is the desolvated compound 1 stable to air? Which is the luminescence of desolvated compound 1 after its exposure to air and humidity?
Answer: Thank you for this good suggestion. We tried to put 1a in liquid water and found a obvious shift to 718 nm. This result was amended in the revised manuscript as “1a was relatively stable toward air and moisture. The emission of 1a shifted to 718 nm when its powder was immersed in liquid water for 20h (Figure S6). We suggested this red shift was caused by the interaction between the Ag10Cu6 cluster center and water molecules. This interaction was weak and lessened when the content of water was decreased. Therefore, a low-energy shoulder could be observed at the emission curve of 1a, which indicated that a little portion of 1a was hydrated by moisture when it was put in open air for days.” (Page 4, Lines 152-158).
Round 2
Reviewer 2 Report
The authors have responded satisfactorily to the referee's suggestions and I am happy to recommend publication of the paper in Molecules.